# Interleukin-1β in Multifactorial Hypertension: Inflammation, Vascular Smooth Muscle Cell and Extracellular Matrix Remodeling, and Non-Coding RNA Regulation

**DOI:** 10.3390/ijms22168639

**Published:** 2021-08-11

**Authors:** Elaina Melton, Hongyu Qiu

**Affiliations:** Center for Molecular and Translational Medicine, Institute for Biomedical Sciences, Georgia State University, Atlanta, GA 30303, USA; emelton@gsu.edu

**Keywords:** interleukins, IL-1β, inflammation, hypertension, non-coding RNA, vascular smooth muscle cell

## Abstract

The biological activities of interleukins, a group of circulating cytokines, are linked to the immuno-pathways involved in many diseases. Mounting evidence suggests that interleukin-1β (IL-1β) plays a significant role in the pathogenesis of various types of hypertension. In this review, we summarized recent findings linking IL-1β to systemic arterial hypertension, pulmonary hypertension, and gestational hypertension. We also outlined the new progress in elucidating the potential mechanisms of IL-1β in hypertension, focusing on it’s regulation in inflammation, vascular smooth muscle cell function, and extracellular remodeling. In addition, we reviewed recent studies that highlight novel findings examining the function of non-coding RNAs in regulating the activity of IL-1β and its associated proteins in the setting of hypertension. The information collected in this review provides new insights into understanding the pathogenesis of hypertension and could lead to the discovery of new anti-hypertensive therapies to combat this highly prevalent disease.

## 1. Introduction

Hypertension, or high blood pressure, is a multifactorial disease related to genetic, environmental, demographic, lifestyle, vascular, and neuroendocrine disorders; and is a serious medical condition that can increase the risk of heart, brain, kidney, and other diseases, thus making it a major cause of premature death worldwide [1,2,3,4,5,6]. Based on the most recent WHO report in 21 May 2021, an estimated 1.13 billion people worldwide have hypertension, and less than 20% of these individuals have the problem under control (https://www.who.int/news-room/fact-sheets/detail/hypertension, accessed on 21 May 2021). Although the terminology and classification of hypertension can vary, the following three major hypertensive disorders are highly recognized as the most significant and intriguing unsolved problems: systemic arterial hypertension (SAH)—the elevation of arterial blood pressure in the vessels that supply oxygenated blood to the body; pulmonary hypertension (PH)—the increased blood pressure within the arteries of the lungs; and gestational hypertension (GH) referring to the high blood pressure developed during the pregnancy. Hypertension can also be classified into primary hypertension (essential hypertension) or secondary hypertension based on the causes. Primary hypertension accounts for about 85–90% of the cases, indicating that a monocausal etiology has not been identified. Despite many years of efforts, hypertension remains one of the most challenging disorders to study and treat due to the complex nature and unknown cause, thus intensifying the need for more investigations to identify the factors or molecular signaling that cause most of these disorders [7]. One of the advancements in this field is the discovery of possible inter-relationships between the various interleukins and the development of high blood pressure [1,8]. Specifically, a growing amount of evidence points to a possible link between hypertension and interleukin-1β (IL-1β), a well-characterized mediator of inflammation.

IL-1β belongs to the IL-1 family of interleukins, a group of circulating cytokines implicated in inflammation and disease [9,10,11,12]. There are 11 members of the IL-1 family expressed by multiple immune cells such as dendritic cells, monocytes, neutrophils, B cells, and T cells. These immune-modulating cytokines inflict inflammatory responses in various cell types and tissues [10]. It has been shown that these interleukins participate in either pro-inflammatory effects, such as IL-1α, IL-1β, IL-18, and IL-36α, IL-36β, IL-36γ, or anti-inflammatory effects, such as IL-1Ra, IL-33, IL-36 Ra, IL-37, and IL-38 [13]. The biological functions of these interleukins have been linked to a plethora of disease processes, including hypertension [11,13]. This group of interleukins bind to their corresponding receptors and generally signal through the myeloid differentiation factor 88 (MyD88)/IL-1 receptor-associated kinase (IRAK)/TNF receptor-associated factor (TRAF) pathway to facilitate downstream upregulation of inflammatory genes through a nuclear factor kappa-light-chain-enhancer of activated B cells (NFkB) dependent mechanism. Specifically, it has been shown that IL-1β binds to the IL-1R1 receptor and its accessory protein IL-1RAcP to facilitate the transcriptional upregulation of multiple genes such as IL-6, IL-17, IFNγ, leading to downstream pro-inflammatory events linked to disease progression and tissue damage [10,11,12]. This activity can be suppressed by the competitive binding of IL-1Ra to IL-1R1 [14].

IL-1β is well recognized as clinically relevant given its high abundance in patients with hypertension [15]. Because of its essential role in inflammation, examination of how IL-1β influences vascular pathology-related changes in hypertension has gained much interest and is currently being explored. For instance, recent reports suggest that IL-1β not only participates in the pro-inflammatory response in the vessels, it also influences vascular smooth muscle cell (VSMC) phenotype and functions and vascular remodeling in multiple types of hypertension through inflammatory-dependent or independent mechanisms [16]. In addition, another new avenue of research in this area is related to the regulation of IL-1β by non-coding RNAs (ncRNAs) in hypertension [17,18]. These new findings also suggest that analogous regulatory mechanisms may be involved in the pathogenesis of various hypertensive disorders.

In this review, we summarized recent work investigating the role of IL-1β in the development of various types of hypertension and the molecular mechanisms involved, mainly focusing on its pro-inflammatory effects on VSMC functions, including VSMC differentiation migration, proliferation, and vascular remodeling. We also discuss the new progress in understanding how ncRNAs control the activity of IL-1β and its associated proteins in the context of hypertension. This new information can potentially improve our understanding of the molecular mechanisms involved in hypertension pathogenesis and foster the development of therapeutic solutions for hypertension-related diseases, furthermore, helping to establish IL-1β as a valid target for hypertension.

## 2. The Role of IL-1β in Hypertension and the Associated Inflammatory Signaling

Over recent years IL-1β has been linked in various types of hypertension such as SAH, PH, and GH [8,9], highlighting the importance of this interleukin in the progression of hypertension. Studies with inhibitors in humans have also provided some advancements in knowledge and proof of concept. Here we summarized the new progress in the study of IL-1β, focusing on three common types of hypertension, e.g., SAH, PH, and GH, in terms of it’s pro-inflammatory effects.

It has been shown that IL-1β plays an essential role in SAH. For example, a few studies have found that patients with essential hypertension displayed high levels of IL-1β in their serum, indicating the pro-hypertensive effects of IL-1β [8,15]. In line with these results, IL-1Ra, the inhibitor of the IL-1β receptor, was also shown to be elevated in hypertension, which was considered to play a role in combating the IL-1β induced inflammation in hypertension [8,19]. In addition, the vaso-constrictive properties of IL-1β were linked to patients with autosomal dominant polycystic kidney disease (ADPKD). Notably, IL-1β was highly correlative to the incidence of hypertension seen in ADPKD. Interestingly, in this study, there was a strong association between the level of IL-1β and hypoxia-inducible factor-1a (HIF-1a) in the blood, which established a link between the hypoxia and the IL-1β pathway in hypertension [20]. A recent study performed on human subjects diagnosed with obesity further supports that the progression of hypertension may be influenced or regulated by the IL-1β pathway. This study investigated the ability of an IL-1R1 receptor Inhibitor (Anakinra) to modulate blood pressure in obese patients. The results showed that Anakinra had a short and long-term effect on systolic blood pressure [21], and the patients administered this inhibitor also exhibited lowered stroke systemic vascular resistance index and peripheral vascular resistance, which was accompanied by a marked increase in the vasodilator Angiotensin 1-7 (Ang 1-7). These results indicated that IL-1 inhibition involves the modulation of the renin-angiotensin-aldosterone system (RAAS), whereby Ang 1-7 induces vasodilation to regulate blood pressure [21]. On the contrary, results from the Canakinumab Anti-Inflammatory Thrombosis Outcome Study (CANTOs) showed that IL-1β might not be linked to hypertension given that hypertension did not improve in patients given the IL-1β inhibitor Canakinumab [22]. These contradicting results have given rise to the debate regarding whether inhibiting IL-1β is beneficial in treating hypertension and therefore needs to be further investigated.

In addition, studies from in vitro, animal models, and in humans also showed that IL-1β is associated with PH. For example, IL-1β was discovered to be a key inflammatory marker present in the serum of PH subjects [23]. Evidence of this strong association was also reported by Soon et al. and others who found that PH patients exhibited high levels of IL-1β [24,25]. Pulmonary vascular pathology has also been linked to proteins involved in the IL-1β pathway. For instance, IL-1R1, the IL-1β receptor, and MyD88, its downstream adaptor protein, are highly expressed in the pulmonary vessels of PH patients as well as the PH mouse model induced by hypoxia [26]. Thus collectively, these reports provide further validation that IL-1β is a crucial component in pro-hypertensive settings.

Mechanistically, loss of function studies following the inhibition of IL-1β with IL-1Ra, showed that depressing its activity lowers pulmonary vascular resistance and reduces changes in vascular morphology in a mouse model of PH [27]. This study corroborates an earlier study showing that inhibition of IL-1β activity by genetically or pharmacologically inhibiting its receptor, IL-1R1 with an IL-1 antagonist Anakinra, prevented both hypoxia-induced PH in mice and monocrotaline-induced PH in rats [26]. Silencing macrophage-specific expression of MyD88, the molecular adaptor protein downstream of IL-1R1 stimulation, also disrupted the progression of PH in mice, leading to the speculation that the IL-1R1/MyD88 interaction is vital for facilitating signaling events that lead to hypertension [26]. In line with these studies, suppressing IL-1R1 with Anakinra has also improved inflammation and cardiovascular outcomes in both human and animal models of PH [25,28,29]. Additionally, the NOD-, LRR- and pyrin domain-containing protein 3 (NLRP3) inflammasome, a key modulator of IL-1β, has been shown to be linked to PH development, thus making it an ideal target for disease treatment [30]. For example, several studies have demonstrated that inhibition of NLRP3 depresses the development of PH in mice and rats, likely by disrupting the downstream signaling events that lead to IL-1β and IL-18 production, release, and subsequent inflammatory reactions [31,32]. Similarly, it has been predicted that deleting necessary components of the NLRP3 inflammasome machinery, such as the apoptosis-associated speck-like adapter protein (ACS), would have a beneficial effect in lowering hypertension in lung arteries [33]. In line with that concept, the work published by Cero demonstrated that PH development was suppressed and the abundance of IL-1β and IL-18 in lung lysates was lowered when ACS was silenced in the hypoxia-induced PH mouse model [34].

Furthermore, plasma and placenta levels of pro-inflammatory IL-1α, IL-1β, and IL-6 were found to be significantly increased in women with GH compared to the normotensive pregnant patients [35]. A human case study done in 2011 corroborates Benyo’s 2001 study showing that IL-6 and IL-1β are higher in pre-eclampsia (PE), a life-threatening type of maternal hypertensive disorder, patients than observed in the control groups [36]. In another human study, with the onset of GH, the IL-1β enriched cytokine profile was paired with higher triglycerides, total cholesterol, and low-density lipoprotein cholesterol. This disruption of normal lipid levels was accompanied by a lower level of high density lipoprotein (HDL), which is classified as cholesterol transporter that has anti-inflammatory properties. This data indicates a possible link between inflammation and cholesterol metabolism in GH [37]. Taken together, it suggests that circulating levels of interleukins, particularly IL-1β, may be a determinant of GH progression in pregnant women.

Reciprocally, Southcombe and colleagues demonstrated a strong association between the anti-inflammatory effects of several IL-1 family cytokine members and PE. Notably, they reported a significant increase in anti-inflammatory molecules soluble IL-1RAcP (sIL-1RAcP), IL-37, and IL-18BP in the blood and placenta samples from PE patients compared to the control group [38]. sIL-1ACP is the truncated form of the IL-1R1 adapter protein IL-1RAcP, which acts as an inhibitor of the IL-1R1 receptor and thus blocks IL-1β activity. The presence of anti-inflammatory sIL1AcP in PE conditions suggests that it may be elevated to alleviate the pro-inflammatory events associated with IL-1β induced inflammation in PE. Inhibition of IL-1β in the context of PE has also been studied recently. Astragaloside IV (AsIV), a known natural anti-inflammatory agent, was found to be beneficial in lowering systolic blood pressure and mRNA levels of IL-1β and IL-6 in a lipopolysaccharide-induced PE rat model [39]. It was determined that AsIV treatment is likely beneficial in reducing PE through a mechanism that involves TL4/NF-kB.

## 3. The Regulation of IL-1β in VSMC Function and ECM Remodeling in Hypertension

VSMCs are major components of the vascular wall and serve to mediate hemodynamic vessel functions, playing a crucial role in regulating blood pressure in physiological conditions and hypertension. Under pathological conditions, VSMCs undergo differentiation, contractile, proliferative, and migratory alterations [40,41,42]. These changes disrupt the function of vessels and can contribute to disease progression [40]. Additionally, the extracellular matrix (ECM), another key component of the vascular wall, which also plays an essential role in vascular function. ECM remodeling has been considered a critical pathological process in the development of hypertension [43,44]. Recently, interleukins’ role in inducing changes in VSMC and ECM has caught much attention. Here we shed some light on recent advances related to IL-1β mediated phenotypic changes in VSMCs occurring during the development of hypertension in both inflammatory dependent and independent mechanisms.

Early reports point to a potential relationship linking inflammation/IL-1β and it’s associated proteins to SAH-related changes in the SMC phenotype [16], suggesting that VSMC becomes more contractile when exposed to IL-1β. Evidence revealed that IL-1β treatment upregulated pro-inflammatory genes, which is mediated through an NFkB-dependent mechanism [45], suggesting that VSMCs can transform into a more pro-inflammatory state in response to IL-1β.

In addition, IL-1β related VSMC migration may also play a significant role in vascular remodeling in SAH. It was shown that IL-1β could promote cell migration in VSMC via a mechanism that involves metalloproteinase-2 (MMP2). Upon inhibition of MMP2, the effect of IL-1β on migration was blunted, thus confirming a relationship between IL-1 β, MMP2, and VSMC migratory pathways [46]. Another study showed that migration was attenuated in VSMCs from SHR rats by inhibiting NLRP3 activation with an antioxidant called Curcumin. It was reported that IL-1β expression and NFkB activation were reduced in the Curcumin-treated cells, further implicating this pathway in VSMC migration [47].

Furthermore, several studies have determined NLRP3 to be involved in VSMC proliferative processes. Indeed, when NLRP3 is perturbed by Toll-like receptor 4 (TLR4) genetic silencing, VSMC proliferation, IL-1β, and blood pressure are reduced in SHR rats. This implies a link between NLRP3, IL1β, and the TLR4 signaling cascade that may be involved in VSMC proliferation [48]. Reciprocally, a study showed that uric acid treatment leads to proliferation of VSMCs and marked upregulation of components of the NLRP3 pathway, including IL-1β, indicating that this proliferation process likely occurs through an NLRP3 dependent pathway [49]. Another study showed that deletion of Caspase 1, a central component in the NLRP3 inflammasome processing pro-IL-1β into active IL-1β, prevents hypoxia-induced PH in Knockout mice. Notably, VSMC proliferation was also significantly reduced in the Caspase 1 deficient mice, indicating that modulating the NLRP3/IL-1β pathway can affect VSMC proliferation [50].

In addition to the changes of VSMC phenotype and functions, ECM remodeling is also a hallmark of hypertension progression and disease. It has been shown that remodeling of the vasculature and the progression of aortic fibrosis was depressed in SHR rats lacking the calcium-sensing receptor (CasR). In the CasR knockout rats, the NLRP3 inflammasome, IL-1β, and collagen production was reduced, indicating a relationship between Ca^+^ influx, IL-1β, and vascular remodeling [51]. Recent reports showed that hypertension development and EMC changes were limited in SHR rats exposed to ischemic conditioning; this phenotype was accompanied by a marked decrease in IL-1β, implying the depressing effects on the ECM was due to the lowered IL-1β [52]. In addition, blocking NLRP3 with fibronectin type III domain-containing 5 (FNDC5) reduced not only oxidative stress, NLRP3 activity, and vascular remodeling in VSMCs but also modulated Amp-activated protein kinase (AMPK)-histone/protein deacetylase Sirtuin1 (SIRT1) activity, a known interaction that plays a role in energy metabolism. This group demonstrated that NLRP3/ IL-1β could influence VSMC function through an AMPK-SIRT1 dependent mechanism influenced by metabolic homeostasis [53].

## 4. Regulation of IL-1β by ncRNAs

Although it has been generally assumed that most genetic information is transacted by proteins via encoding mRNAs, recent evidence suggests that the majority of the genomes of mammals is transcribed into ncRNA that does not encode a protein, including microRNAs (miRNAs), small interfering RNAs (siRNAs), PIWI-interacting RNAs (piRNAs) and various classes of long ncRNAs (lncRNAs). Despite the fact that many aspects of their biology remain to be understood, increasing evidence has unveiled a slew of influential roles of the ncRNA as transcriptional and post-transcriptional regulators; thus, they are expected to be associated with cellular dysfunction and disease [54,55,56,57]. So far, ncRNA is linked to many cardiovascular diseases, and this growing body of work is nicely summarized by Zhang and others [58,59]. Given that inflammation is a common factor in vascular pathologies, considerable research linking ncRNAs to the inflammatory responses involved in hypertension has been established [60,61,62]. Consistent with this idea, several studies have identified hypertension-related miRNAs such as miR-106b-5p, microRNA-124, and miR-129, which possibly influence hypertension by modulating inflammation [63,64,65,66,67]. Based on the high relevance and association of IL-1β to hypertension, recent studies have led to the discovery of several ncRNAs that target this gene.

Several ncRNAs that regulate immune-related processes have been explored in the pathology governing SAH. Most recently, miR-1929-3p was found to reduce SAH by targeting the inflammasome pathway, a major contributor of IL-1β production and release. In this study, they found that mice overexpressing miRNA-1929-3p exhibited less vascular remodeling and NLRP3 inflammasome activation. This likely occurred by miRNA-1929-3p mediated suppression of the endothelin A receptor (ETRA) which has been shown to be critical in endothelin-mediated blood pressure changes and inflammasome activation [18,68]. Similarly, it was found that miR155-5p limits oxidative stress and IL-1β in the SHR rat model [69]. Thus, providing evidence that this miRNA is involved in preventing pro-hypertensive cellular responses. In a separate study, a lncRNA uc.48+ was named an important element in the development of SAH. Its inhibitory actions are characterized by preventing high blood pressure and lowering IL-1β levels. Upon genetic knockdown of this ncRNA with a siRNA against uc.48+, blood pressure (BP) and serum levels of IL-1β were depressed in the high-fat diet and stretch-induced diabetic mouse model. Since P2X7R (purinergic type 2 receptors) expression was significantly decreased after silencing uc.48, it was speculated that changes in BP might be due to lowered P2X7 activity [70]. Taken together, the effectiveness of ncRNAs targeting IL-1β in preventing hypertension further suggests that this pathway is a likely target.

In addition to the effects on SAH, a recent study showed that overexpression of lncRNA Carbamoyl-phosphate synthetase 1-Intronic Transcript (CPS1-IT) was associated with reduced PH and lowered IL-1β in a mouse model of sleep apnea, likely through a mechanism involving the downregulation of HIF1 [17]. miR-340-5p was also shown to have a preventative role in the development of PH. It was demonstrated that miR-340-5p inversely correlated with IL-1β and IL-6 levels in acute pulmonary embolism patients with PH. miRNA340-5p also dampened the level of SMC proliferation and migration, likely by suppressing the activities of IL-1β and IL-6 [53].

Furthermore, a recent study showed that the interaction between a lncRNA named MALAT1, and a microRNA identified as miR-150-5p, can lead to the progression of GH in mice. Increased blood pressure in this mouse model of GH was likely caused by the upregulation of a vasoconstrictor called Endothelian-1 (ET-1). It was suggested that the increase in ET-1 likely occurs as a result of MALAT1 blocking the repressive activity of miR-150-5p which targets IL-1β. This study links miR-150-5p to SMC contraction, IL-1β, and hypertension [71].

While strides in understanding how miRNA and lncRNA modulate the immune-pathological pathways in hypertensive disease, some attention has been focused on the impacts on VSMC phenotypic switch and dysfunction during hypertension pathogenesis. For example, by using microarray analysis to explore genetic changes in response to inflammation, it has been found that miR-25 was decreased when VSMCs were exposed to IL-1β. miR-25 was found to be a regulator of Kruppel like factor 4 (KLF4), a transcription factor that mediates VSMC specific genes such as MyH11. This led to the suggestion that inflammatory factors, such as IL-1β, can block the repression effect of KLF4 by inhibiting miR-25 [72]. In this way, IL-1β not only induces inflammation but also contributes to the SMC changes associated with hypertensive pathology by regulating miRNA.

## 5. Conclusions and Future Directions

In summary, there is a growing amount of evidence to suggest that IL-1β is involved in the pathogenesis of various types of hypertension, indicating analogous regulatory mechanisms may be involved in vascular dysfunction despite the distinct initial causes. The collected information highlighted here suggest that IL-1β participates in the development of hypertension through its mediated inflammatory signaling and by regulating VSMC function, and ECM remodeling. Furthermore, as summarized in Figure 1 and Table 1, several studies provide evidence showing that ncRNAs modulate IL-1β during the progression of hypertension, indicating a comprehensive regulation among the ncRNA, inflammation, and VSMC function. The studies reviewed here bring new insight into how IL-1β impacts the pathogenesis of hypertension and may yield promising avenues in exploring new targets for the treatment of various hypertension conditions.

Although it is accepted that IL-1β is highly associated with the development of hypertension, its effects and underlying regulatory mechanisms are far from fully understood, and some reports are still controversial. Future studies are needed to investigate the essential role of IL-1β by genetic and pharmacological inhibition of IL-1β to confirm its effects in physiological and pathological conditions. In addition, more research is required to explore the interaction of IL-1β and its associated proteins and their inter-network with ncRNAs. Furthermore, since the interleukins synergically or competitively work together during many diseases, the interaction of IL-1β with other IL-1 members needs to be further investigated. Notably, new studies involving pro-inflammatory interleukin IL-18 and anti-inflammatory interleukins IL-37 and IL-38 have provided interesting new findings related to hypertension prevention. Thus, finding ways to regulate the pro-and anti-inflammatory activities of these immune modulators and the ncRNAs that regulate them is of interest and provides potential for more research areas in hypertension-related diseases.

It is well established the molecular mechanisms associated with obesity can increase the risk for cardiovascular diseases such as hypertension. Recently ncRNAs have been recognized as potential regulators and biomarkers for obesity. Interestingly, human studies found that the miRNA profiles of obese individuals were different from those exhibited by overweight and normal-weight subjects. Specifically, miR-34a was abundantly elevated in obese subjects compared to overweight individuals [73]. Distinct ncRNAs characteristic of morbidly obese adults and overweight children has also been identified, thus further implicating ncRNAs in the pathology of obesity [74,75]. Since low-grade inflammation is a hallmark of obesity progression, it is reasonable to suggest that it may be the cause of the alteration of ncRNAs observed in obese individuals [76]. Indeed, the miRNA profiles of adipocytes from obese subjects were altered in response to lipopolysaccharide (LPS) treatment, which therefore suggests that LPS induced inflammation can lead to modifications in miRNA [77]. On the contrary, the idea that low-grade inflammation is a consequence of ncRNA regulation rather than the cause has also been tested. miRNAs, namely miRNA-17, miRNA-20a, and miRNA-106a, have been shown to play a role in promoting the infiltration of immune cells into adipose tissue, and thus contribute to low-grade inflammation that accompanies obesity, which further implicates ncRNAs as drivers of inflammation regulation [78]. Moreover, in the future, it would be interesting to understand how ncRNA regulates IL-1β functions at the intersection of inflammation and obesity, particularly in the context of hypertension.

Given the effects of exercise on disease, its relation to ncRNA regulation in disease has been extensively explored [79,80,81]. In the context of hypertension, physical activity has been proven to be beneficial in resolving complications related to the pathogenesis of hypertension [82,83]. Evidence suggests that these beneficial effects may be in part due to changes in ncRNA expression in response to physical activity [84,85]. For example, a study found that SHR rats exposed to exercise for ten weeks were rescued from hypertension. The improved blood pressure was accompanied by altered expression of ncRNAs that targeted several molecular mediators of HT (i.e., Angiotensin-converting enzyme (ACE) and Angiotensin-converting enzyme 2 (ACE2), and AngII type 1 receptor (AT1R) [64,86]. This study suggested that exercise prevents hypertension through a mechanism that involves ncRNA-mediated regulation of the Renin-Angiotensin pathway. Given its proven positive effect on hypertension and cardiovascular health, it would be beneficial to study the effects of physical activity on the changes in exercise-related miRNAs that control the activity of IL-1β.

The impact of diet and the use of food supplements on miRNA regulation of inflammation has also been considered a necessary area of study [87]. For example, in a 2019 study, obese individuals exposed to a ketogenic diet exhibited miRNA levels similar to lean subjects [88]. This phenotype was also paralleled by changes in the expression level of antioxidant and anti-inflammatory genes. In line with this work, a study examining the effects of using omega-3 supplements in combination with the Ketogenic diet improved cardiovascular outcomes in overweight individuals [89]. In addition, a recent clinical case study provided evidence that implementing the ketogenic diet is beneficial in preventing pulmonary hypertension [90]. Furthermore, these studies collectively give new insight into the understanding of how diet can modulate inflammation and hypertension. Whether the ketogenic diet or other supplements promote the suppression of IL-1β activity by miRNAs is an unknown area of interest.

## Figures and Tables

**Figure 1 ijms-22-08639-f001:**
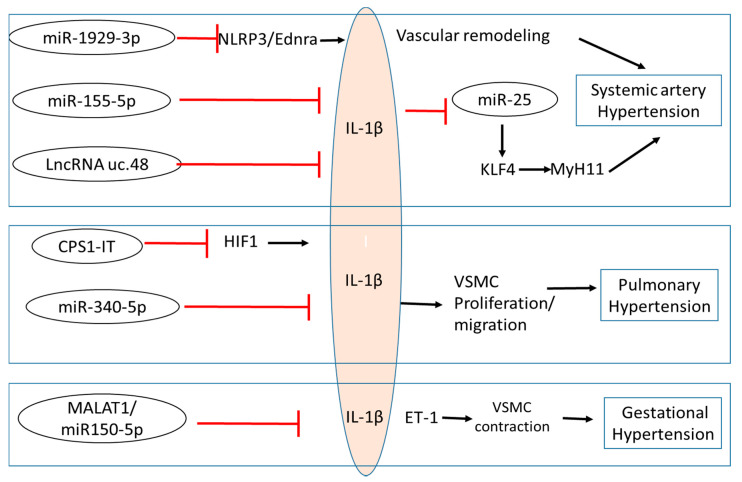
Non-coding RNA regulation of IL-1β and its associated signaling in various hypertension conditions. Several ncRNAs function to block the pro-inflammatory actions of IL-1β in the setting of systemic hypertension, pulmonary hypertension, and gestational hypertension. IL-1β also inhibits miR-25 and is subsequently involved in the regulation of blood pressure. NLRP3: NLR family pyrin domain containing 3; ETRA: endothelin A receptor; KLF4: Kruppel like factor 4; HIF1: Hypoxia-inducible factors; ET1: Endothelin 1; CPS1-IT: CPS1 Intronic Transcript 1; MALAT 1: metastasis-associated lung adenocarcinoma transcript 1; VSMC: vascular smooth muscle cell.

**Table 1 ijms-22-08639-t001:** Summary of Biological Action of ncRNA targeting the IL-1β pathway.

ncRNA	Model	Function	Role in HT	References
miR-1929-3p	Murine Cytomegalovirus Induced SAH	- Inhibits vascular remodeling - Limits NLRP3 inflammasome activation	Anti-hypertensive	Zhou et al., 2020 [18] Wang et al., 2020 [68]
miR-155-5p	Rat WKY and SHR VSMCs ± Angiotension II treatment	- reduces ACE expression - limits oxidative stress - lowers IL-1β production - reduces VSMC migration	Anti-hypertensive	Wu et al., 2020 [69]
LncRNA uc.48+	Murine High Fat Diet Type 2 Diabetic model associated SAH	- lowers blood pressure - reduces IL-1β - downregulates P2X7 expression	Pro-hypertensive	Wu et al., 2020 [70]
CPS1-IT	Murine Obstructive sleep apnea associated PH	- reduces IL-1β - downregulates of HIF1	Anti-hypertensive	Zhang et al., 2019 [17]
miR-340-5p	Human Acute pulmonary embolism associated PH	- dampens SMC proliferation and migration - suppresses IL-1β and IL-6 actions	Anti-hypertensive	Zhou et al., 2020 [53]
MALAT1/miR150-5p	Murine model of GH	- upregulates vasoconstrictor Endothelian-1 - MALAT1 disrupts miR150-50 mediated repression of IL-1β	Pro-hypertensive	Ou et al., 2020 [71]

## Data Availability

Not applicable.

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
