# Peer review of "Interleukin-1β in Multifactorial Hypertension: Inflammation, Vascular Smooth Muscle Cell and Extracellular Matrix Remodeling, and Non-Coding RNA Regulation"

_ijms, 2021, doi:10.3390/ijms22168639_

Round 1
Reviewer 1 Report
The work is well written and well covers the state of the art on the subject.
I suggest expanding the section regarding miRNA, especially those related to inflammation and possible modulation.
But above all, it would be interesting to have a distinction, in human studies, between obese and overweight subjects as the increase in them could be due to low-grade inflammation.
In addition, the effect of physical activity could also be analyzed, which is known to have a positive effect on hypertension.
Likewise, the action of the diet and the use of food supplements could / should have an effect on the inflammatory state, so it should be considered an action in this direction; highlighting how diet could modulate inflammation (and miRNA), particularly ketogenic diet.
Reviewer 2 Report
The review manuscript entitled “Interleukin-1β in multifactorial hypertension: inflammation, vascular smooth muscle cell and extracellular matrix remodeling, and non-coding RNA regulation” by Dr. Melton and Dr. Qui provides a detailed overview of the current knowledge concerning the role of Interleukin-1β (IL-1β) in the pathogenesis of hypertension, including systemic arterial, pulmonary and gestational hypertension. Details on the relationship between IL-1β and non-coding RNAs upon hypertension onset are provided, as well. Overall, the text is interesting, well written and easy to follow. It improves our knowledge regarding the role of IL-1β as well as IL-1β/ncRNAs axis on hypertension pathogenesis. The ms is in general well written, detailed and organized.
While I recommend this review manuscript for publication, I have few observations.
Main comments
1. This ms presents several typo (word spacing) errors. For instance, lines 44, “specifically and “a growing” are separated by too many spaces. Other lines: 27, 63, 72, 94, 98, 99, 119, 124 as well as many others. Please revise the text accordingly
2. Few topics throughout the text are lacking in supporting references, I have suggested some refs
3. A table summarizing the main ncRNAs targeting IL-1β transcript should be helpful for the reader
Minor comments
Lines 37-42. These sentences are lacking in supporting references. This topic has been extensively reviewed here PMID: 25832858
Lines 142-145 This sentence is lacking in supporting references
Line 149 “IL-1a, IL-1β” --> please uniform the style, as IL-1a should be IL-1α (as reported in line 52, for instance)
Line 182 Plese remove “(REF)”
Line 237. “…increasing evidence has unveiled a slew of influential roles of the ncRNA as transcriptional and post-transcriptional regulators”--> This sentence is lacking in supporting references. A detailed information of the regulative function of ncRNAs/lncRNAs at both transcriptional and post-transcriptional level, in physiological conditions, can be found here (PMID: 33898434).
Line 254 P2X7 should be P2X7R
Line 325 Please revise the authors’ names of the ref n.2
Lines 267-268. It is unclear why miR-150-5p is described as a lncRNA
Round 2
Reviewer 1 Report
I think the authors have made enough improvement for the work to be published